# Privacy-Preserving MRI Data Harmonization for Black-box Models

## Abstract

In MRI, variations in scan parameters, sequence, or hardware can lead to discrepancies in image appearance, even for the same subject. These inconsistencies, known as domain shifts, can hinder image analysis and degrade the performance of deep learning models trained on data from specific source domains. MRI *harmonization* aims to address these issues by aligning target domain images to the source images while preserving anatomical structures. However, most existing harmonization methods require access to both source and target domain data, making data sharing essential and potentially compromising the data privacy that is critical in medical domain. To address this, we propose **BboxHarmony**, the first harmonization framework tailored for black-box settings, where requires neither data sharing nor access to downstream task model parameters. Our approach estimates the source domain style by searching the manifold of MRI domain style constructed via a disentanglement-based generator using Bayesian optimization guided by black-box model performance. We evaluated our method on brain tissue segmentation task across multiple institutes and demonstrated that it effectively harmonizes target images into source images, leading to improved downstream task performance of a black-box model. By enabling harmonization under strict data-sharing and model-access constraints, BboxHarmony opens an uncharted area of privacy-preserving harmonization in clinical applications.

## 1 Introduction

Magnetic resonance imaging (MRI) is a prevalent medical imaging modality, serving a pivotal role in disease diagnosis, monitoring, and treatment planning. Recent advances in deep learning have significantly enhanced automated MRI image analysis, facilitating more accurate and robust approaches. However, one of the major obstacles for deploying these models in a real-world clinical setting is the domain shift problem: MRI data exhibits substantial variations across different vendors, scanners, and scan parameters even when imaging the same subject (Cai et al., 2021). Consequently, a model trained on one domain (referred to a source domain), often demonstrates significantly degraded performance when applied to data from the other domains (referred to a target domain). Here, we adopt the terminology from the domain adaptation literature, where model is trained on *source* domain while the *target* domain refers to unseen domain.

Several approaches have been proposed to address this domain shift problem. Traditional transfer learning through fine-tuning utilizes paired images and labels from the target domain to adapt pre-trained models (Tajbakhsh et al., 2016). Domain adaptation techniques offer an alternative by aligning feature distributions between source and target (Ben-David et al., 2006; Long et al., 2015), but these methods frequently fail to preserve essential anatomical information — a non-negotiable requirement in medical applications. Furthermore, fine-tuning and domain adaptation approaches depend on access to the model parameters (Tab. 1), which can leak sensitive information about the data used to train the model through model inversion attacks (Haim et al., 2022; Yang et al., 2025).

Harmonization has emerged as a promising strategy that aligns images from target domains to match a specific source domain, removing domain-specific biases while preserving biological information such as anatomical structure. Importantly, harmonization operates without requiring access to the parameters of pre-trained models, instead functioning by mapping target data distributions toward the source domain. Conventional harmonization methods span from traditional approaches like his-

Table 1: Comparison between our task formulation, existing domain shift reduction methods, and *Black-box harmonization* in terms of *i)* data sharing requirements and *ii)* access to pre-trained task model parameter. An additional column, *black-box constraint*, indicates whether both requirements are absent. Methods satisfying this condition are marked with ✓, while those requiring either data sharing or parameter access are marked with ✗.

| Setting | Data sharing | Model parameter accessibility | Black-box constraints |
|---|---|---|---|
| Fine-tuning (Tajbakhsh et al., 2016) | not required | required | ✗ |
| Domain adaptation (Ben-David et al., 2006; Long et al., 2015) | required | required | ✗ |
| Conventional harmonization (Dewey et al., 2019; Modanwal et al., 2020; Liu et al., 2021a; Jeong et al., 2023; Beizaee et al., 2025; Roca et al., 2025) | required | not required | ✗ |
| **Black-box harmonization (ours)** | not required | not required | ✓ |

togram matching and statistical normalization to advanced deep learning-based techniques. For example, DeepHarmony (Dewey et al., 2019) uses paired data from traveling subjects scanned across domains, while unsupervised methods like CycleGAN (Zhu et al., 2017) eliminate this need but still require access to both source and target domain data (Modanwal et al., 2020; Liu et al., 2021a). More recently, target-free harmonization methods (Jeong et al., 2023; Beizaee et al., 2025) have been introduced. Despite recent advances, a key challenge remains that most existing harmonization methods require data sharing or exportation for model development (Tab. 1). This compromises data privacy issues, which is critical in the medical domain.

A practical examples including various domain shift reduction scenarios illustrates in Fig. 1. If the hospital has access to a sufficiently large labeled dataset, it can train its own task network (Fig. 1a). In cases where labeled dataset is small, transfer learning of a model trained on a large dataset may be employed (Fig. 1b), but this typically requires data sharing, which can raise data privacy concerns. Conventional harmonization methods offer an alternative by training a harmonization network to align their own data to the source domain data (Fig. 1c), yet they still depend on access to both source and target domain data. However, in many clinical settings under strict regulations (e.g., HIPAA, GDPR), deep learning models are often deployed as privacy-preserving black-box (e.g., via APIs or fixed software), which restrict access to internal parameters and prevent data sharing (Price, 2018; Price & Nicholson, 2014). Consequently, existing domain gap reduction methods cannot be applied in such black-box environments. This motivates us to consider a more realistic scenario, where a hospital performs harmonization using only its own data, without sharing it externally (Fig. 1d).

To address this challenge, we proposed **BboxHarmony**, the first MRI data harmonization framework designed for black-box models under strict data sharing constraints. Our approach requires only the target domain data and operates without any access source domain data. This approach marks a fundamental shift from existing harmonization methods to privacy-preserving method. BboxHarmony employs a disentanglement-based MRI style generator capable of synthesizing a diverse spectrum of MRI styles while preserving anatomical information. Then, we search the latent space of the generator to estimate unknown source domain style guided by optimal performance from the black-box model. Given the high cost of querying the black-box model and the high dimensionality of generator's latent space, which requires capturing rich domain-specific variations, we employ a Bayesian optimization that enables efficient search. Our key contributions are as follows:

- We present the first harmonization method specifically designed for privacy-preserving black-box settings, addressing a critical requirement in clinical environment.

- We develop a disentanglement-based generative framework that enables diverse style manipulation while preserving important anatomical information of MRI images.

- Our method demonstrates the efficacy of Bayesian optimization for navigating complex latent style spaces using only black-box performance feedback.

## 2 RELATED WORKS

### 2.1 MRI HARMONIZATION

The harmonization of MR images from different sources has become a crucial technique for mitigating domain shifts. Early approaches relied on techniques such as histogram matching (Shinohara

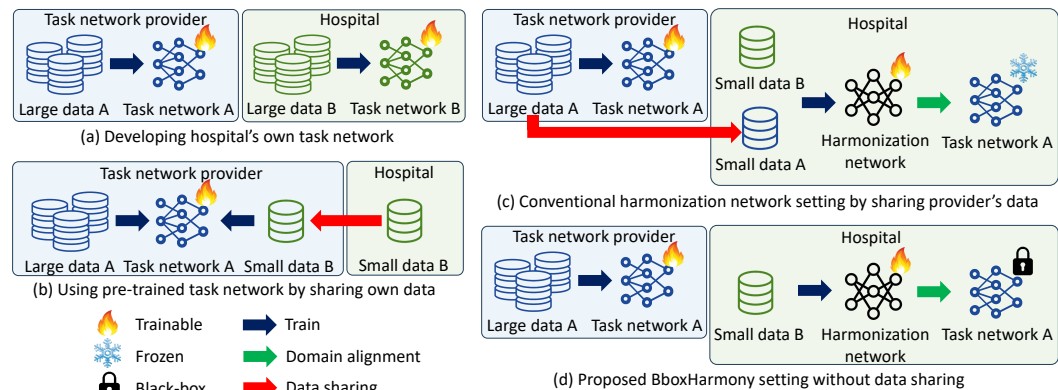

Figure 1: Overview of domain adaptation and harmonization settings in clinical environment. (a) A hospital with a large dataset can train its own task network. (b) With limited data, it can adapt a pre-trained network via transfer learning, but requiring data sharing. (c) Conventional harmonization enables using pre-trained network without fine-tuning but requires sharing of task network's training data. (d) Our proposed method trains a harmonization network using only small amount of in-house data, without data sharing or access to the task network's parameters, addressing practical constraints like data privacy and scarcity in medical field.

et al., 2014; Nyúl et al., 2000; Papamakarios et al., 2021) and statistical normalization (Fortin et al., 2017; Pomponio et al., 2020; Shinohara et al., 2017), which primarily adjust contrast and intensity. With the advent of deep learning, more sophisticated harmonization methods have emerged. Supervised approaches such as DeepHarmony (Dewey et al., 2019) and unsupervised style transfer methods (Modanwal et al., 2020; Liu et al., 2021a; Roca et al., 2025) have shown promising results, but they all require access to both source and target domain datasets, introducing practical data privacy challenges. More recently, target-free harmonization methods (Beizaee et al., 2025; Jeong et al., 2023) have been introduced, reducing data acquisition costs by eliminating the need for target domain data. However, these approaches are often feasible only from the model developer's perspective, where the source data used in model training are available, and thus remain impractical for data-holding hospitals that lack such access.

## 2.2 DISENTANGLED REPRESENTATION LEARNING

Disentangled representation learning has emerged as a powerful paradigm for separating domain-invariant content from domain-specific style (Bengio et al., 2013; Gatys et al., 2016). In medical imaging, such disentanglement has been employed to isolate anatomical structures from varying domain styles, allowing controlled image manipulation while preserving biologically relevant information (Pei et al., 2021; Yang et al., 2019). A primary application includes cross-modality synthesis (Reaungamornrat et al., 2022; Wang & Zheng, 2021), data augmentation (Gu et al., 2023; Cai et al., 2025). Disentanglement learning has also been incorporated into MRI harmonization (Zuo et al., 2021a;b; Liu & Yap, 2024; Dewey et al., 2020). By learning distinct latent representations for anatomical structure and imaging contrast, methods like CALAMITI (Zuo et al., 2021a;b) enable fine-grained control over harmonized image attributes, successfully preserving content while modifying only style during harmonization.

## 2.3 BAYESIAN OPTIMIZATION

Bayesian optimization (BO) is a framework for optimizing objective functions that are expensive to evaluate (Brochu et al., 2010). It leverages a probabilistic surrogate model, typically a Gaussian Process (GP), to approximate the objective function and quantify the associated uncertainty (Snoek et al., 2012; Frazier, 2018). With the trained surrogate model, an acquisition function guides the next evaluation point selection by balancing exploration and exploitation (Jones et al., 1998; Kushner, 1964; Srinivas et al., 2010). This allows efficient optimization when evaluation of the objective function is costly. Recent work has extended BO to high-dimensional problems by using dimensionality reduction or structured kernels to enhance optimization performance (de Freitas & Wang, 2013; Kandasamy et al., 2015; Moriconi et al., 2020; Letham et al., 2020; Nayebi et al., 2019;

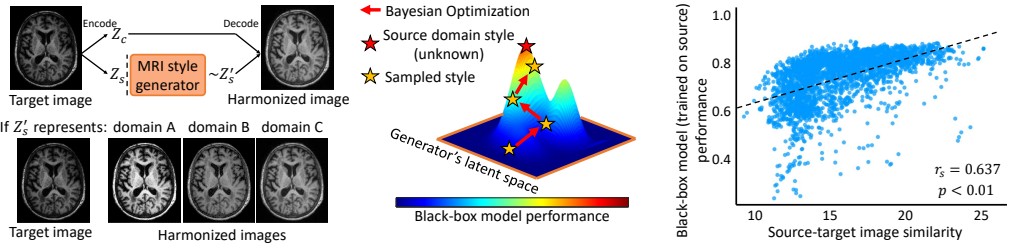

(a) Overall pipeline of BboxHarmony    (b) Estimating unknown source domain style    (c) Analysis of black-box performance and domain difference

Figure 2: (a) We design a generator trained via disentanglement learning to preserve anatomical content $z_c$ while generating diverse MRI domain styles $z_s$. This enables harmonization by synthesizing images in various domain styles. enabling harmonization by sampling $z'_s$ corresponding to specific domains. (b) To estimate the unknown source domain style in a black-box setting, Bayesian optimization explores the generator's latent space guided by black-box model performance. (c) Empirical analysis shows a positive correlation between Dice score (black-box performance) and image SSIM (source-target domain image similarity), supporting our assumption that higher black-box performance reflects greater similarity between input and source domain.

Wilson et al., 2016). Moreover, BO has also been explored in domain adaptation, specifically for optimizing hyperparameters that control the domain adaptation process (Muratore et al., 2021; Li & He, 2020). While BO has been applied to various domain adaptation tasks, its application to MRI harmonization remains underexplored.

## 3 METHOD

### 3.1 MOTIVATION: BLACK-BOX PERFORMANCE AS A PROXY FOR DOMAIN ESTIMATION

In a black-box harmonization scenario, we cannot access to information of source domain, hindering application of any of the previously proposed harmonization approaches. In this scenario, the only observable indication of the unknown source domain is the performance of the black-box model itself, which is trained on the data from source domain. This constraint led us to hypothesize that performance degradation of the black-box model may be related to the magnitude of domain shift between the source and target distributions. Formally, let $\mathcal{T}$ denote the target domain, $\mathcal{S}$ represent the unknown source domain, and $\mathbf{x}^t \in \mathcal{T}$ be an image from the target domain. We denote the black-box model trained on $\mathcal{S}$ as $M_{\text{bbox}}$, and the task performance of $M_{\text{bbox}}$ on an input image $\mathbf{x}$ as $P(\mathbf{x}; M_{\text{bbox}})$. We assume that the black-box task performance on a target image $\mathbf{x}^t$ can be modeled in relation to the performance on the source domain, $P(\mathbf{x}; M_{\text{bbox}})$, and the domain shift $\Delta(\mathbf{x}^t, \mathbf{x})$ between $\mathbf{x}^t$ and $\mathbf{x}$ with a task-dependent sensitivity coefficient $\alpha$:

$$P(\mathbf{x}^t; M_{\text{bbox}}) \approx P(\mathbf{x}; M_{\text{bbox}}) - \alpha \cdot \Delta(\mathbf{x}^t, \mathbf{x}). \tag{1}$$

We empirically verified Eq. (1) through a controlled pilot experiment using traveling subject data from four MRI domains (one for source, and the others for targets). For the black-box model $M_{\text{bbox}}$, we trained a brain tissue segmentation network with the source domain. For each pair of the domains, we computed the image similarity between source and target domain images using SSIM, and evaluated the black-box model performance with the Dice score (Dice, 1945). The results in Fig. 2c revealed a positive correlation between the source and target image similarity and the black-box performance.

These results led to the key insight that the black-box task performance implicitly encodes information about the domain shift magnitude between the target and unknown source domain distribution. By treating the black-box task performance as a proxy for domain alignment quality, we can guide the harmonization process without direct access to the source domain. Our approach transforms harmonization into an optimization problem:

$$\mathbf{x}^* = \arg\max_{\mathbf{x} \in \mathcal{G}} P(\mathbf{x}; M_{\text{bbox}}). \tag{2}$$

Here, $\mathcal{G}$ represents the manifold that represents the characteristics of diverse MRI domains, and $\mathbf{x}^*$ denotes the harmonized image that best approximates the unknown source domain in terms of black-box task performance. This formulation enables the search for an unknown source domain through

iterative optimization guided by black-box network performance, establishing a novel paradigm for black-box harmonization. To facilitate this, we need to construct the manifold space $\mathcal{G}$, and explore it efficiently. The following sections describe the design of BboxHarmony, which meets these requirements through an MRI style generator via disentanglement and Bayesian optimization.

## 3.2 GENERATING A MANIFOLD REPRESENTING STYLES OF DIVERSE MRI DOMAINS

**Disentanglement-based Generator.** To construct the manifold that captures the characteristics of diverse MRI domains, we adopt disentangled representation learning to separate domain-specific style from domain-invariant content (Fig 2a). Here, we define domain-invariant content as the underlying anatomical structures in MRI images, while domain-variant style as image appearance factors that contribute to inter-domain variation, such as contrast, blur, and noise (Kushol et al., 2023).

Our generator adopts a content-style disentanglement framework (Gatys et al., 2016), composed of a content encoder ($\mathcal{E}_c$), style encoder ($\mathcal{E}_s$), and decoder ($\mathcal{D}$). Given an input MRI image $\mathbf{x}$, the content and style encoders extract a content vector $\mathbf{z}_c = \mathcal{E}_c(\mathbf{x})$ and a style vector $\mathbf{z}_s = \mathcal{E}_s(\mathbf{x})$, respectively. These vectors are concatenated and passed to the decoder to reconstruct the image $\hat{\mathbf{x}}$:

$$\hat{\mathbf{x}} = \mathcal{D}(\mathbf{z}_c, \mathbf{z}_s) = \mathcal{D}(\mathcal{E}_c(\mathbf{x}), \mathcal{E}_s(\mathbf{x})). \tag{3}$$

For generation of synthetic MRI image with diverse style, the decoder takes the content vector of the input MRI image and a randomly sampled style vector from the Gaussian distribution as:

$$\mathbf{x}' = M_G(\mathbf{z}'_{\mathbf{s}}; \mathbf{z}_{\mathbf{c}}) = \mathcal{D}(\mathcal{E}_c(\mathbf{x}), \mathbf{z}'_{\mathbf{s}}), \quad \mathbf{z}'_{\mathbf{s}} \sim \mathcal{N}(\mathbf{0}, \mathbf{I}), \tag{4}$$

where $M_G$ is the generator and $\mathbf{x}'$ is a generated image from a randomly sampled style $\mathbf{z}'_{\mathbf{s}}$.

For training, we construct a paired dataset consists of an original MRI image and its synthetically perturbed counterpart. Perturbations include random combinations of contrast adjustment, blurring, and noise injection, reflecting common targets of the MRI image variability (Kushol et al., 2023). These perturbations alter the style while preserving anatomical structure, providing natural supervision for content-style disentangling. The detailed training objectives are provided in the Appendix A.

**Strategies to Increase Generator Expressiveness.** The use of perturbed pairs allows the style encoder to learn from synthetic variations. To further enhance the expressiveness of the style space, we incorporate MRI images from three different scanners during training. Details of training dataset are described in Sec. 4. For each domain, we generate perturbed image pairs and train the generator with a shared style encoder, embedding all images into a unified latent space. These domains exhibit realistic style differences arising from variations in scan parameters and hardware, therefore allowing the style encoder to generalize across a wider range of MRI image styles. While exhaustive coverage of all domain styles is not guaranteed, this multi-domain training scheme encourages the model to capture a broader spectrum of plausible MRI styles beyond those represented by synthetic perturbations alone. It is important to note that no source domain data is used during the generator training. The generator serves as the foundation of our harmonization framework, enabling searching an unknown source domain style from the latent space of it via BO, as described in Sec. 3.3.

## 3.3 BAYESIAN OPTIMIZATION FOR ESTIMATING UNKNOWN SOURCE DOMAIN STYLE

To discover the best approximation of the unknown source domain style vector, we adopt BO from two complementary perspectives: (i) efficiency in querying a black-box model during inference, and (ii) scalability in exploring the high-dimensional style space of our generator.

**Problem Formulation.** In our scenarios, we define an objective function $f(\cdot)$ that maps each sampled MRI style vector to the observed black-box model performance. This function reflects how closely a given style vector approximates the unknown source domain (Fig. 2b). Specifically, we evaluate $f(\cdot)$ by averaging the black-box performance over a batch of target-domain content images:

$$f(\mathbf{z}_{\mathbf{s}}) = \frac{1}{|X_{\text{train}}|} \sum_{\mathbf{x} \in X_{\text{train}}} P\big(M_{\text{bbox}}\big[M_G(\mathbf{z}'_{\mathbf{s}}; \mathbf{z}_{\mathbf{c}})\big]\big), \tag{5}$$

where $X_{\text{train}}$ denotes a set of original input MR images from multiple style samples. Conclusively, we aim to solve $\mathbf{z}'_{\mathbf{s}\star} = \arg\max_{\mathbf{z}'_{\mathbf{s}} \in \mathcal{G}} f(\mathbf{z}'_{\mathbf{s}})$, identifying a style vector that produces harmonized images most aligned with the unknown source domain.

(a) Disentanglement results on traveling subjects  (b) Disentanglement results on unpaired subjects

Figure 3: **Qualitative evaluation of disentanglement.** Each column shows a content image (top), synthesized output (middle), and style reference (bottom). (a) Paired setting with traveling subjects: same anatomy, different style. (b) Unpaired setting: different anatomy and style. In both cases, outputs reflect the reference style while preserving anatomical structure.

**Optimization Procedure.** To estimate the optimal MRI style vector $\mathbf{z}'_{\mathbf{s}\star}$, we implement BO with a GP surrogate model, initially trained on random style vectors and their black-box performance. After initialization of the GP model, we iteratively select new candidate style vectors and evaluate using GP-UCB acquisition function (Srinivas et al., 2010). Then, the most promising candidate is selected for querying the black-box model. The candidate and its corresponding black-box performance are then incorporated into the GP training set to update the surrogate model. This process is repeated until convergence. This strategy enables efficient optimization under limited query budgets by focusing evaluations on informative style vectors. The complete optimization process is outlined in **Algorithm 1**.

---

**Algorithm 1** BO search for source-like style vector

---

**Require:** generator $M_G$, black-box $M_{\text{bbox}}$, training images $X$, init $N_0$, objective function $f(\cdot)$, budget $T$, trade-off $\beta$

1: *(init)* Sample $N_0$ style vectors $\mathbf{z}'^{(i)}_{\mathbf{s}} \sim \mathcal{N}(0, I)$, $i \in [0:N_0-1]$, and set $\mathcal{B} = \{(\mathbf{z}'^{(i)}_{\mathbf{s}}, f(\mathbf{z}'^{(i)}_{\mathbf{s}}))\}$
2: **for** $t = 1$ **to** $T$ **do**
3:     Fit GP surrogate on $\mathcal{B}$
       *// GP-UCB acquisition function*
4:     Select $\mathbf{z}'^{(t)}_{\mathbf{s}} \leftarrow \arg\max_{\mathbf{z}'_{\mathbf{s}}} [\mu_{t-1}(\mathbf{z}'_{\mathbf{s}}) + \beta\sigma_{t-1}(\mathbf{z}'_{\mathbf{s}})]$
5:     Evaluate $y_t = f(\mathbf{z}'^{(t)}_{\mathbf{s}})$
6:     $\mathcal{B} \leftarrow \mathcal{B} \cup \{(\mathbf{z}'^{(t)}_{\mathbf{s}}, y_t)\}$
7: **end for**
8: **return** $\mathbf{z}'^{\star}_{\mathbf{s}} = \arg\max_{(\mathbf{z}'_{\mathbf{s}}, y) \in \mathcal{B}} y$

---

## 4 EXPERIMENTAL RESULTS

**Experimental Setup.** For the experiments, we performed brain tissue segmentation as a downstream task of a black-box model. For the black-box network architecture, a U-Net (Ronneberger et al., 2015) was used. We utilized T1-weighted images from the OASIS-3 dataset (LaMontagne et al., 2019), which consists of images from several vendors and scanners. The ground-truth labels of brain tissue masks were generated using FSL FAST (Jenkinson et al., 2002). Total of five Siemens scanners from Siemens were employed for our experiments, where four were designated as target domains (Domain A, B, C, and D), and the other as the source domain. For the generator training, we excluded target domain D to assess whether the generator can perform harmonization on a domain it has not encountered during training. To further assess the generalization capability of our approach, we also evaluated on MRI data from vendors not used in training (e.g., GE and Philips), thereby testing the robustness of the method across scanner manufacturers beyond Siemens (See Appendix G). To train the black-box segmentation model, 1,380 subjects from the source domain were used, while BboxHarmony only utilized five subjects per target domain. Each subject had 50 slices. All images were resampled to a uniform voxel size ($1.2 \times 1.2 \times 1$ mm$^3$) and underwent percentile normalization at the slice level. The harmonization network was trained for 2D slices. More detailed data information is in the Appendix B.

**Evaluation of the Disentanglement-Based Generator.** To evaluate our generator for disentanglement, a synthesized image was generated from the content and style vectors from the content and reference style images, respectively. This evaluation was conducted in two settings: a paired traveling subject setting with identical anatomy but different styles, and an unpaired subject setting with differing anatomy and style. As shown in Fig. 3, our generator successfully preserved anatomical structures of the content image while adapting the style from the style image in both settings. PSNR

and SSIM between the output and the style images in the paired setting supported effectiveness of our generator's disentanglement (See the Appendix C for the results).

To visualize the coverage of generated images in a embedding space, we trained an auxiliary MRI domain classifier and applied t-SNE to its intermediate features from real and generated images. As shown in Fig. 4, real images from the four domains (A, B, C, and D) formed domain-specific clusters, while generated images were more broadly distributed, indicating successful coverage of diverse MRI styles.

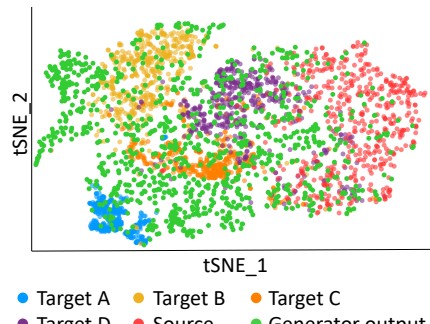

**Evaluation of Source Domain Style Estimation with Bayesian Optimization.** We evaluated whether BO can efficiently identify MRI style vectors that align with an unknown source domain. To validate its effectiveness in navigating the generator's high-dimensional style space, we compared BO against random search (Bergstra & Bengio, 2012). We tracked the black-box model's performance with the sampled style vectors by the two methods over time. To assess whether the optimization also translates into improved

Figure 4: t-SNE visualization based on an MRI domain classifier. Real MRI target domains form distinct clusters, while images generated from our model are more widely dispersed even covering parts of unseen source domain.

harmonization quality, SSIM between harmonized target and paired source images from validation dataset is also tracked. As shown in Fig. 5, BO reached higher-performing regions faster than random search, both in task performance and image similarity.

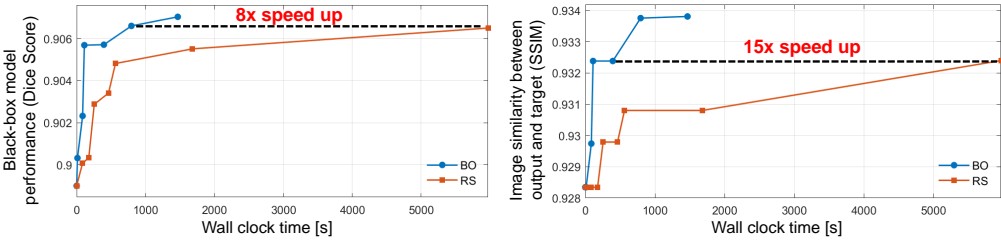

(a) Black-box model performance over wall-clock time     (b) Output-target image similarity over wall-clock time

Figure 5: **Bayesian optimization (BO, blue) versus random search (RS, orange).** (a) Black-box model performance (Dice Score) as a function of wall-clock time. (b) Image similarity (SSIM) between output and source over the same time span. BO reaches higher performance earlier than RS, illustrating its faster convergence in both Black-box model performance and image similarity.

**Evaluation of Harmonization via Inferred Source Domain Style.** To evaluate BboxHarmony, we applied the estimated source style to target images from traveling subjects and compared the results with corresponding source images. As baselines, we included manual perturbation (random combinations of contrast, blur, and noise tuned for the target domain) and prior harmonization methods including DeepHarmony (Dewey et al., 2019), style transfer (Liu et al., 2021a), BlindHarmony (Jeong et al., 2023), Harmonizing flows (Beizaee et al., 2025), and IGUANe (Roca et al., 2025). PSNR and SSIM were used for quantitative comparison. As shown in Tab. 2, all harmonization methods improved the image similarities except for BlindHarmony, which requires substantial source dataset to learn data distribution. BboxHarmony outperformed the manual perturbation, while DeepHarmony achieved the highest similarity thanks to the using of paired training data. Fig. 6 presents qualitative comparisons across the methods. The results of other domains are in Appendix E. The manual perturbation resulted in visible discrepancies from the source image, indicating its limited ability to account for complex domain shifts. DeepHarmony achieved close visual alignment with the source. However, it produced overly smoothed outputs, which is a known artifact of a U-Net-based architecture. Our proposed method successfully harmonized target image without accessing to the source data.

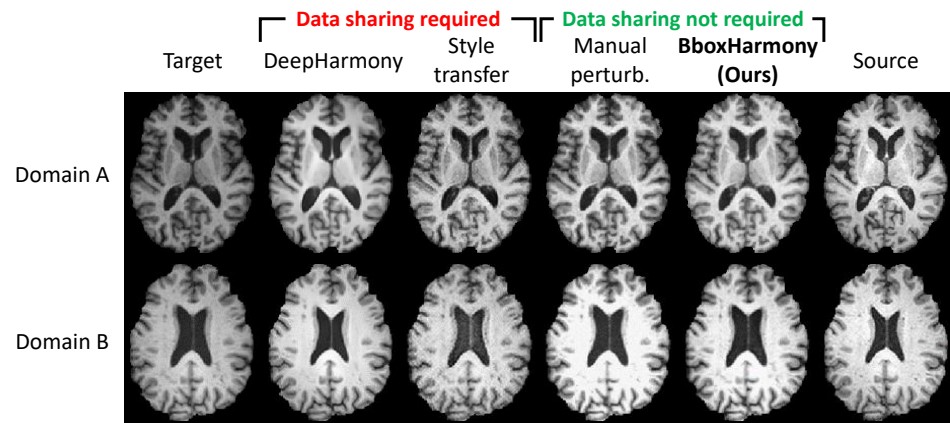

Figure 6: Visual comparison of harmonization results across two target domains (A and B) using different methods (see Appendix E for results on other domains). Methods marked in red require domain sharing, while those in green–including ours do not.

Tab. 3 and Fig. 7 summarize segmentation performance following the application of various harmonization methods (the results of other domains are in Appendix E). Without harmonization, the black-box model experienced a substantial performance drop due to domain shift. Most harmonization approaches mitigated this issue, with the exception of DeepHarmony. Despite utilizing paired source-target data, DeepHarmony tended to produce overly blurred outputs, likely due to its architectural design, which ultimately degraded segmentation performance. These results demonstrate that the our method improves the performance of the black-box model on unseen target domain.

Table 2: Quantitative metrics of image similarity (PSNR and SSIM) between source and target images before (no harmony) and after harmonization using different methods (DeepHarmony, Style transfer, BlindHarmony, Harmonizing flows, IGUANe, manual perturbation, and BboxHarmony) across four target domains. Notably, BboxHarmony achieves performance comparable to source data-required methods without using source data.

| Methods | Data sharing | Domain A | | Domain B | | Domain C | | Domain D | |
|---|---|---|---|---|---|---|---|---|---|
| | | PSNR↑ | SSIM↑ | PSNR↑ | SSIM↑ | PSNR↑ | SSIM↑ | PSNR↑ | SSIM↑ |
| No harmony | - | 17.8 ±1.2 | 0.883 ±0.022 | 13.9 ±1.7 | 0.844 ±0.061 | 15.7 ±2.6 | 0.907 ±0.045 | 18.4 ±2.3 | 0.909 ±0.042 |
| DeepHarmony (Dewey et al., 2019) | required | 23.6 ±1.8 | 0.936 ±0.019 | 19.0 ±1.6 | 0.887 ±0.053 | 20.2 ±1.9 | 0.933 ±0.035 | 20.8 ±2.2 | 0.924 ±0.035 |
| Style transfer (Liu et al., 2021a) | required | 20.5 ±1.3 | 0.915 ±0.012 | 17.4 ±1.6 | 0.871 ±0.020 | 18.6 ±1.6 | 0.903 ±0.016 | 19.7 ±2.0 | 0.910 ±0.018 |
| BlindHarmony (Jeong et al., 2023) | required | 10.1 ±1.6 | 0.755 ±0.047 | 11.3 ±1.8 | 0.799 ±0.079 | 12.2 ±3.0 | 0.855 ±0.081 | 11.9 ±2.9 | 0.831 ±0.074 |
| Harmonizing flows (Beizaee et al., 2025) | required | 18.6 ±1.1 | 0.889 ±0.018 | 16.6 ±2.1 | 0.865 ±0.064 | 18.5 ±2.1 | 0.923 ±0.042 | 18.4 ±3.0 | 0.911 ±0.043 |
| IGUANe (Roca et al., 2025) | required | 18.6 ±1.9 | 0.908 ±0.023 | 17.0 ±2.0 | 0.861 ±0.064 | 18.6 ±1.6 | 0.922 ±0.045 | 19.4 ±2.4 | 0.911 ±0.044 |
| Manual perturbation | not required | 19.8 ±1.5 | 0.912 ±0.022 | 16.9 ±2.2 | 0.823 ±0.067 | 17.7 ±2.1 | 0.915 ±0.045 | 19.0 ±2.4 | 0.909 ±0.050 |
| **BboxHarmony (ours)** | not required | 20.2 ±1.3 | 0.923 ±0.019 | 17.5 ±1.7 | 0.869 ±0.062 | 18.4 ±1.9 | 0.922 ±0.044 | 19.3 ±2.4 | 0.911 ±0.043 |

## 5 DISCUSSION

In this paper, we proposed BboxHarmony, a novel privacy-preserving harmonization method designed for a black-box setting where both data sharing and access to model parameter are inaccessible. By leveraging a disentanglement-based generator, our approach successfully separates domain-invariant anatomical content from domain-variant imaging style enabling to only convert the MRI style component to another domain (Fig. 3). Notably, our generator demonstrated the ability to synthesize images that more closely resemble unseen source domain styles when provided with their style representations, despite having no access to those domains during training. This observation suggests a potential for generalization beyond the training domains (see the Appendix C).

BboxHarmony benefits from the expressive capacity of the learned MRI style manifold. Our generator captures domain-specific styles, as evidenced by a higher quantitative metric (Tab. 2), enabling effective harmonization across diverse MRI domains. Leveraging this expressiveness, BO efficiently estimates the source domain style solely through black-box performance feedback, without requiring access to source domain data or model parameters (Fig. 5, 6; Tab. 2). This black-box compatibility marks a notable advancement over prior harmonization methods, enabling improved downstream segmentation performance (Fig. 7 and Tab. 3). Improved image similarity and downstream task per-

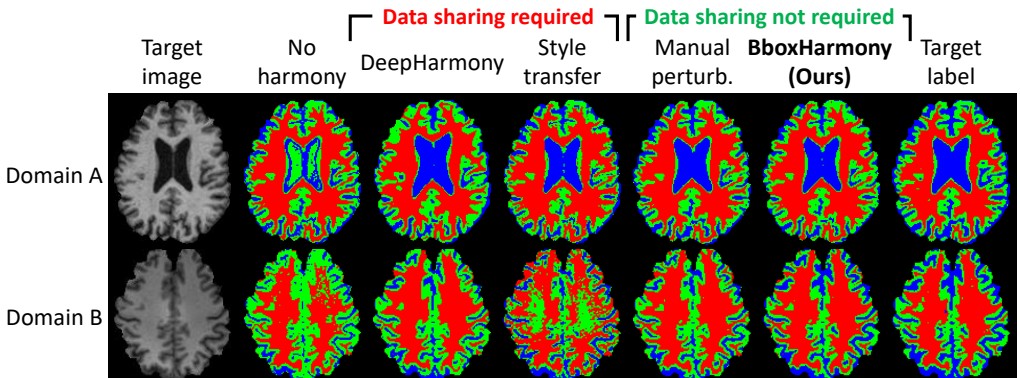

Figure 7: Brain tissue segmentation results on target images from two domains (A and B) applying different harmonization methods (results on other domains are in Appendix E). Without harmonization (no harmony), performance drops due to domain shift, while harmonization methods generally improves performance. BboxHarmony successfully segments brain tissues, which demonstrate that our method enables the black-box models achieve better performance on an unseen target domain.

formance on data acquired from unseen vendors (e.g., Philips and GE), which were not used during training, also demonstrate a degree of generalizability of our method (see Appendix G).

Although our method does not require data sharing, it requires a small amount of the labeled target data. We explored training task networks directly on target data without harmonization, but observed performance degradation when small amount of data were used due to overfitting (see Appendix F). In clinical settings, such labels may be scarce or costly to obtain. Therefore, these findings highlight the continued importance of harmonization under practical constraints.

Despite these strengths, BboxHarmony has several limitations. First, our experimental evaluation is restricted to a limited set of domains drawn from the training dataset, and may not fully capture the diversity of real-world MRI protocols. Moreover, our framework lacks an explicit mechanism to constrain or quantify the coverage of the learned MRI style manifold. Future work should evaluate its robustness on a broader range of imaging conditions, including different acquisition sequences (e.g., T2-weighted) and modalities (e.g., CT, PET). Lastly, our experiments primarily involved healthy subjects. It remains unclear whether the generator preserves clinically relevant features when applied to pathological data, such as lesions. Validation on diverse and pathological datasets is crucial to ensure the clinical reliability of BboxHarmony in real-world diagnostic applications.

Table 3: IoU and Dice scores for brain tissue segmentation before (no harmony) and after harmonization using various methods (DeepHarmony, Style transfer, BlindHarmony, Harmonizing flows, IGUANe, Manual perturbation, BboxHarmony) across four target domains (Domain A, B, C, D).

| Methods | Data sharing | Domain A | | Domain B | | Domain C | | Domain D | |
|---|---|---|---|---|---|---|---|---|---|
| | | IoU↑ | Dice↑ | IoU↑ | Dice↑ | IoU↑ | Dice↑ | IoU↑ | Dice↑ |
| No harmony | - | $0.711_{\pm 0.034}$ | $0.830_{\pm 0.023}$ | $0.750_{\pm 0.067}$ | $0.852_{\pm 0.049}$ | $0.772_{\pm 0.076}$ | $0.861_{\pm 0.064}$ | $0.822_{\pm 0.033}$ | $0.900_{\pm 0.023}$ |
| DeepHarmony (Dewey et al., 2019) | required | $0.790_{\pm 0.030}$ | $0.882_{\pm 0.019}$ | $0.651_{\pm 0.058}$ | $0.784_{\pm 0.053}$ | $0.710_{\pm 0.056}$ | $0.822_{\pm 0.054}$ | $0.704_{\pm 0.064}$ | $0.823_{\pm 0.046}$ |
| Style transfer (Liu et al., 2021a) | required | $0.751_{\pm 0.035}$ | $0.856_{\pm 0.024}$ | $0.749_{\pm 0.051}$ | $0.853_{\pm 0.038}$ | $0.720_{\pm 0.063}$ | $0.828_{\pm 0.059}$ | $0.775_{\pm 0.036}$ | $0.871_{\pm 0.027}$ |
| BlindHarmony (Jeong et al., 2023) | required | $0.448_{\pm 0.135}$ | $0.588_{\pm 0.131}$ | $0.637_{\pm 0.082}$ | $0.763_{\pm 0.072}$ | $0.635_{\pm 0.095}$ | $0.759_{\pm 0.077}$ | $0.658_{\pm 0.105}$ | $0.781_{\pm 0.082}$ |
| Harmonizing flows (Beizaee et al., 2025) | required | $0.790_{\pm 0.038}$ | $0.881_{\pm 0.024}$ | $0.787_{\pm 0.053}$ | $0.877_{\pm 0.038}$ | $0.774_{\pm 0.069}$ | $0.863_{\pm 0.061}$ | $0.804_{\pm 0.034}$ | $0.889_{\pm 0.024}$ |
| IGUANe (Roca et al., 2025) | required | $0.806_{\pm 0.037}$ | $0.890_{\pm 0.024}$ | $0.806_{\pm 0.054}$ | $0.890_{\pm 0.040}$ | $0.799_{\pm 0.065}$ | $0.879_{\pm 0.059}$ | $0.827_{\pm 0.029}$ | $0.903_{\pm 0.020}$ |
| Manual perturbation | not required | $0.764_{\pm 0.057}$ | $0.864_{\pm 0.040}$ | $0.804_{\pm 0.085}$ | $0.886_{\pm 0.076}$ | $0.792_{\pm 0.084}$ | $0.873_{\pm 0.072}$ | $0.822_{\pm 0.036}$ | $0.900_{\pm 0.026}$ |
| **BboxHarmony (ours)** | not required | $\mathbf{0.830}_{\pm 0.024}$ | $\mathbf{0.906}_{\pm 0.023}$ | $\mathbf{0.825}_{\pm 0.034}$ | $\mathbf{0.902}_{\pm 0.023}$ | $\mathbf{0.805}_{\pm 0.068}$ | $\mathbf{0.884}_{\pm 0.060}$ | $\mathbf{0.830}_{\pm 0.033}$ | $\mathbf{0.905}_{\pm 0.023}$ |

# 6 CONCLUSION

We presented BboxHarmony, the first MRI harmonization framework designed for privacy-preserving black-box settings, which operates without data sharing nor access to the downstream task network parameters. Our method leverages disentangled representation learning to construct an MRI style manifold that captures domain-specific variations while preserving anatomical content. Using Bayesian Optimization, BboxHarmony efficiently estimates the source domain style within this latent space and harmonizes target images successfully. This approach significantly broadens the applicability of harmonization in real-world clinical environments under strict privacy constraints.

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

# Appendix

## A  IMPLEMENTATION DETAILS

**Disentanglement-based generator.**    The architecture of our generator is illustrated in Fig. S1. The content encoder consists of three convolutional layers followed by instance normalization (Ulyanov et al., 2017) and four residual blocks (He et al., 2016). The style encoder includes three convolutional layers, a global average pooling layer, and a fully connected layer, producing a 32-dimensional style vector. The decoder comprises three upsampling and convolutional layers. To effectively inject style information, we integrate residual blocks with adaptive instance normalization (AdaIN) (Huang & Belongie, 2017) during decoding.

To train the generator, we use a pair of MRI images $\mathbf{x}$ and its perturbed image $\tilde{\mathbf{x}}$, where perturbations are applied to encourage content-style disentanglement. Specifically, $\tilde{\mathbf{x}}$ is generated by applying random combinations of three perturbations in `opencv` (Bradski, 2000), which are contrast adjustment with $\alpha \in [0.5, 1.5]$ and $\beta \in [-20, 60]$, Gaussian blurring with $\sigma \in [0, 0.7]$, and Gaussian noise injection with $\sigma \in [0, 0.01]$. The overall training objective ($\mathcal{L}_{total}$) is a weighted sum of reconstruction loss ($\mathcal{L}_{recon}$), disentanglement loss ($\mathcal{L}_{disent}$), adversarial loss ($\mathcal{L}_{adv}$), and KL-divergence loss ($\mathcal{L}_{KL}$):

$$\mathcal{L}_{total} = \lambda_{recon}\mathcal{L}_{recon} + \lambda_{disent}\mathcal{L}_{disent} + \lambda_{adv}\mathcal{L}_{adv} + \lambda_{KL}\mathcal{L}_{KL}, \tag{S1}$$

$$\mathcal{L}_{recon} = \|\mathbf{x} - \mathcal{D}(\mathbf{z_c}, \mathbf{z_s})\|_1 + \|\mathbf{x} - \mathcal{D}(\tilde{\mathbf{z_c}}, \mathbf{z_s})\|_1 + \|\mathbf{x} - \mathcal{D}(\mathcal{E}_c(\mathcal{D}(\tilde{\mathbf{z_c}}, \mathbf{z_s})), \mathbf{z_s})\|_1, \tag{S2}$$

$$\mathcal{L}_{disent} = \|\mathbf{z_c} - \tilde{\mathbf{z_c}}\|_1 + \|\mathbf{z_c} - \mathcal{E}_c(\mathcal{D}(\mathbf{z_c}, \tilde{\mathbf{z_s}}))\|_1 + \|\mathbf{z_s} - \mathcal{E}_s(\mathcal{D}(\tilde{\mathbf{z_c}}, \mathbf{z_s}))\|_1, \tag{S3}$$

$$\mathcal{L}_{adv} = -\left[\log \mathrm{Dis}(\mathbf{x}) + \log[1 - \mathrm{Dis}(\mathcal{D}(\mathbf{z_c}, \tilde{\mathbf{z_s}}))] + \log \mathrm{Dis}(\tilde{\mathbf{x}}) + \log[1 - \mathrm{Dis}(\mathcal{D}(\tilde{\mathbf{z_c}}, \mathbf{z_s}))]\right], \tag{S4}$$

$$\mathcal{L}_{KL} = D_{\mathrm{KL}}(\mathbf{z_s} \,\|\, \mathcal{N}(0,1)) + D_{\mathrm{KL}}(\mathcal{E}_s(\mathcal{D}(\tilde{\mathbf{z_c}}, \mathbf{z_s})) \,\|\, \mathcal{N}(0,1)), \tag{S5}$$

where $\lambda_{recon}$, $\lambda_{disent}$, $\lambda_{adv}$, and $\lambda_{KL}$ are weights for reconstruction, disentanglement, adversarial, and KL-divergence losses. $\mathcal{E}_c$, $\mathcal{E}_s$, and $\mathcal{D}$ represent content, style encoder, and decoder. The encoders extract a content vector $\mathbf{z}_c = \mathcal{E}_c(\mathbf{x})$, $\tilde{\mathbf{z}}_c = \mathcal{E}_c(\tilde{\mathbf{x}})$ and a style vector $\mathbf{z}_s = \mathcal{E}_s(\mathbf{x})$, $\tilde{\mathbf{z}}_s = \mathcal{E}_s(\tilde{\mathbf{x}})$, which are recombined by $\mathcal{D}$. Additionally, $\mathrm{Dis}(\cdot)$ is the discriminator to provide adversarial feedback.

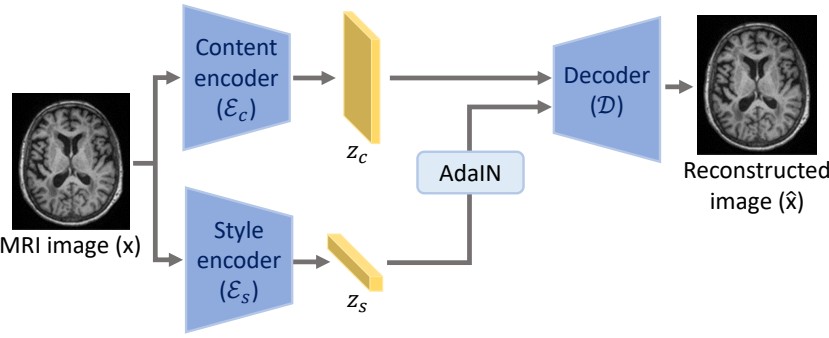

Figure S1: The architecture of the disentanglement-based generator.

**Bayesian optimization for harmonization.**    To implement Bayesian optimization (BO), we model the black-box objective $f(\cdot)$ in Eq. (5) with an exact Gaussian Process (GP) in `gpytorch` (Gardner et al., 2018). We adopt an automatic-relevance-determination radial basis function kernel (Rasmussen & Williams, 2006), customized for our 32-dimensional candidate style vector, $\mathbf{z_s'}, \mathbf{z_s''} \in \mathbb{R}^{32}$ as follows:

$$k(\mathbf{z_s'}, \mathbf{z_s''}) = \sigma^2 \exp\left(-\frac{1}{2} \sum_{d=1}^{32} \frac{(z_d' - z_d'')^2}{\ell_d^2}\right), \tag{S6}$$

where $z_d'$, $z_d''$ respectively denoting the $d$-th coordinate of two style vectors. The dimension-specific length-scales $\ell_d$ enable the GP to attenuate the influence of irrelevant style directions.

We initially train the GP with 100 random style vectors drawn from the generator manifold $\mathcal{G}$ and fit parameters of GP model for 50 iterations with the Adam optimizer (Kingma, 2014) (learning rate 0.1) by maximizing the marginal log-likelihood. At each BO iteration $t$, we sample a candidate set randomly $\{\mathbf{z}_{\mathbf{s}}'^{(t,j)}\}_{j=1}^{100} \subset \mathcal{G}$ and choose the next sample via GP-UCB (Srinivas et al., 2010):

$$\mathbf{z}_{\mathbf{s}}'^{(t)} = \arg \max_{1 \leq j \leq 100} \big[ \mu_{t-1}(\mathbf{z}_{\mathbf{s}}'^{(t,j)}) + \beta \, \sigma_{t-1}(\mathbf{z}_{\mathbf{s}}'^{(t,j)}) \big], \tag{S7}$$

where $\mu_{t-1}$ and $\sigma_{t-1}$ are the GP posterior mean and standard deviation, and $\beta$ balances exploration and exploitation. The new observation is then appended to the training data for GP model, and the GP model is re-optimized before the next step. We repeat this loop for 100 iterations and finally select the style vector, $\mathbf{z}_{\mathbf{s}\star}'$, yielding the highest black-box performance, as follows:

$$\mathbf{z}_{\mathbf{s}\star}' = \arg \max_{t \in \{0,\ldots,99\}} f\big(\mathbf{z}_{\mathbf{s}}'^{(t)}\big). \tag{S8}$$

**Compute time and retargets.** All experiments were run on a single NVIDIA L40S GPU. Training the disentanglement-based generator takes about 35 hours and 32 GB of GPU memory, whereas each sampling with Bayesian optimization takes about 240 seconds and 3.7 GB of GPU memory.

**Code availability.** The target code has been submitted separately as part of the Supplementary material. We will release the full code publicly upon acceptance of the paper. The code for the generator is adapted from MUNIT (Huang et al., 2018)[1] with modifications.

## B EXPERIMENTS SETUP

### B.1 DATASET DESCRIPTION FOR EXPERIMENTS

For BboxHarmony training and evaluation, we used the OASIS-3 dataset (LaMontagne et al., 2019). The source domain was set as the Siemens TIM Trio 3T MR scanner. Four other scanners were used as target domains: Siemens Sonata 1.5T (Domain A), Siemens Vision 1.5T (Domain B), Siemens Magnetom Vida 3T (Domain C), and Siemens BioGraph mMR 3T (Domain D). To standardize resolution, all images were resampled to $1.2 \times 1.2 \times 1$ mm$^3$ and 50 top slices per scan were selected. Acquisition scan parameter details are provided in Tab. S1. For generator training, we used 360 subjects across domains A, B, and C. Note that Domain D was excluded from training and used for evaluation to assess the generator's generalization ability (see Appendix C). For Bayesian optimization-based harmonization, only five labeled subjects from each target domain were utilized. The black-box segmentation network was trained on 1,380 subjects from the source domain.

Table S1: Data descriptions of five domains in OASIS-3 dataset.

| Methods | source domain | Domain A | Domain B | Domain C | Domain D |
|---|---|---|---|---|---|
| Manufacturer | Siemens | Siemens | Siemens | Siemens | Siemens |
| Scanner | TIM Trio | Sonata | Vision | Magnetom Vida | BioGraph mMR |
| Magnetic field strength (T) | 3 | 1.5 | 1.5 | 3 | 3 |
| Matrix size | $176 \times 256 \times 256$ | $160 \times 256 \times 256$ | $128 \times 256 \times 256$ | $176 \times 240 \times 256$ | $176 \times 240 \times 256$ |
| TR/TI (s) | 2.4/1 | 1.9/1.1 | 9.7/unknown | 2.3/unknown | 2.3/0.9 |
| TE (ms) | 3.2 | 3.9 | 4.0 | 3.0 | 3.0 |
| Flip angle(°) | 8 | 15 | 10 | 9 | 9 |

### B.2 COMPARISON METHODS SETUP.

To evaluate the performance of BboxHarmony, we compared it against both a manual perturbation approach and previous deep learning-based harmonization methods (Fig. 6, Fig. 7, Tab. 2, and Tab. 3).

**Manual perturbation.** This baseline applies a combination of random perturbations, including contrast adjustment, blurring, and noise injection, optimized individually for each target domain. Specifically, we randomly applied perturbations to the training set of each target domain over 100 iterations and selected the parameter set that yielded the highest black-box model performance.

---

[1]https://github.com/NVlabs/MUNIT

**Previous Harmonization Methods.** We also compared our method to two representative previous deep learning-based harmonization approaches: DeepHarmony (Dewey et al., 2019) and style transfer method (Modanwal et al., 2020; Liu et al., 2021a). While these methods require access to the source domain, which is feasible in black-box scenarios, the comparison demonstrates how effectively BboxHarmony operates even without any access to the source domain. DeepHarmony was trained using paired traveling subject data from each paired source-target domain, while the style transfer method used unpaired source domain data for training. Both methods were trained on five target domain subjects, which is consistent with BboxHarmony.

## C  ADDITIONAL ANALYSIS OF DISENTANGLEMENT-BASED GENERATOR

We conducted additional experiments to assess whether our proposed generator effectively disentangles anatomical content and style representations across MRI domains. To verify disentanglement, we tested on source-target paired datasets. For each pair, the target domain image was used to extract the content vector, while the source domain image provided the reference style vector. The decoder then synthesized an output image from these two latent vectors. If disentanglement is successful, the synthesized output should exhibit high visual similarity to the reference style image, preserving the original anatomical structure. This process was performed across all four target domains. Qualitative results confirmed that the outputs resembled the style references (Fig. S2), and quantitative evaluation using PSNR and SSIM showed improved similarity compared to the original target images (Tab. S2). Notably, even for domain D, which was excluded during generator training (see Appendix B.1), the results suggest that the generator may generalize beyond the training domains.

To further examine the latent space of the generator, we performed interpolation and extrapolation between style vectors extracted from different MRI images. This experiment was conducted both within the target domains A, B, and C, which were employed for the generator training, and between source and target domains not seen during training. The results showed continuous changes in image appearance while preserving anatomical structure, indicating successful disentanglement of content and style (Liu et al., 2021b) (Fig. S4). This generator enables the synthesis of MRI images with diverse styles, where each style can be viewed as representing a different domain. Fig. S3 illustrates various generated images by combining randomly sampled style vectors with a fixed content vector from the original image indicated by the red box. Notably, the generator is capable of producing images that vary in brightness, contrast, noise level, and blur (Fig. S5).

Table S2: Quantitative similarity (PSNR, SSIM) between synthesized outputs and style reference images across four target domains. Synthesized outputs were generated by combining content vectors from target images with style vectors extracted from paired source domain images.

|  | Domain A | | Domain B | | Domain C | | Domain D | |
|---|---|---|---|---|---|---|---|---|
|  | PSNR↑ | SSIM↑ | PSNR↑ | SSIM↑ | PSNR↑ | SSIM↑ | PSNR↑ | SSIM↑ |
| target | 17.8 | 0.883 | 13.9 | 0.844 | 15.7 | 0.907 | 18.4 | 0.909 |
| Synthesized output | **21.1** | **0.923** | **18.1** | **0.866** | **20.2** | **0.924** | **20.5** | **0.912** |

| Domain A | Domain B | Domain C | Domain D |
| --- | --- | --- | --- |

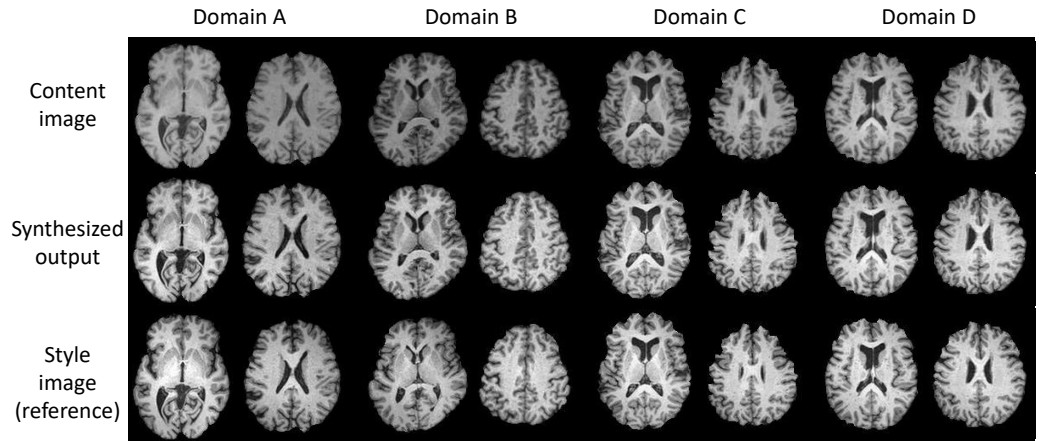

Figure S2: Qualitative evaluation of the generator's disentanglement capability. For each source-target image pair, the target image provided the content representation, and the source image provided the reference style representation. The synthesized outputs resemble the style images while preserving anatomical structure from the content images, demonstrating effective disentanglement.

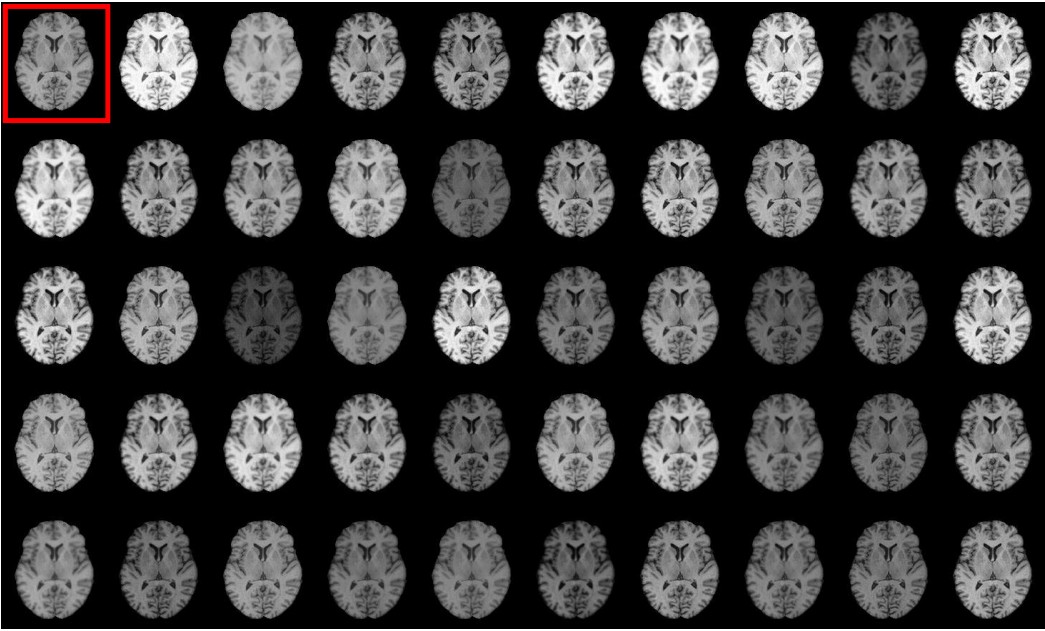

Figure S3: MRI images generated by the disentanglement-based generator. The image marked with the red box is the original image, and the others are generated by replacing its style vector with randomly sampled style vectors, preserving anatomical structure while varying image appearance.

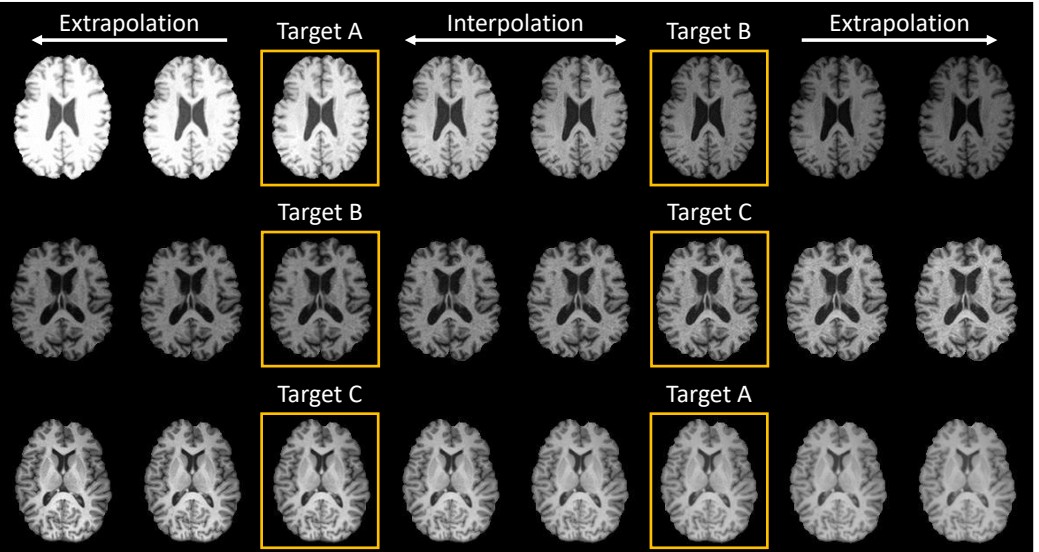

(a) Style interpolation and extrapolation between targets domains

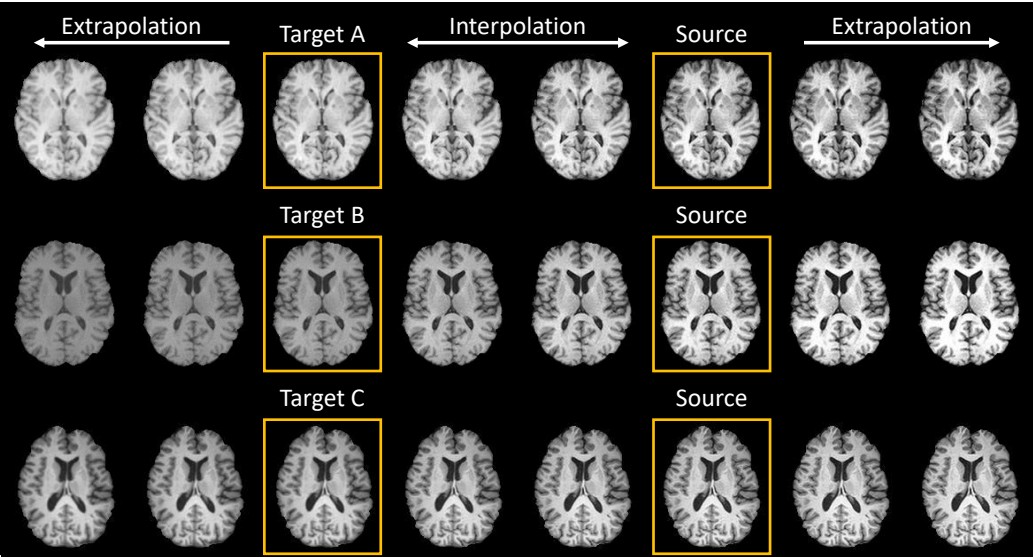

(b) Style interpolation and extrapolation between targets and unseen source domain

Figure S4: Interpolation and extrapolation in the style latent space. Style vectors extracted from two different images were interpolated and extrapolated to generate outputs. (a) shows results from style vectors within target domains used during disentanglement-based generator training, while (b) shows results between target domains and an unseen source domain. The synthesized images show smooth transitions in appearance while maintaining consistent anatomical structure, demonstrating effective disentanglement of the MRI image's content and style.

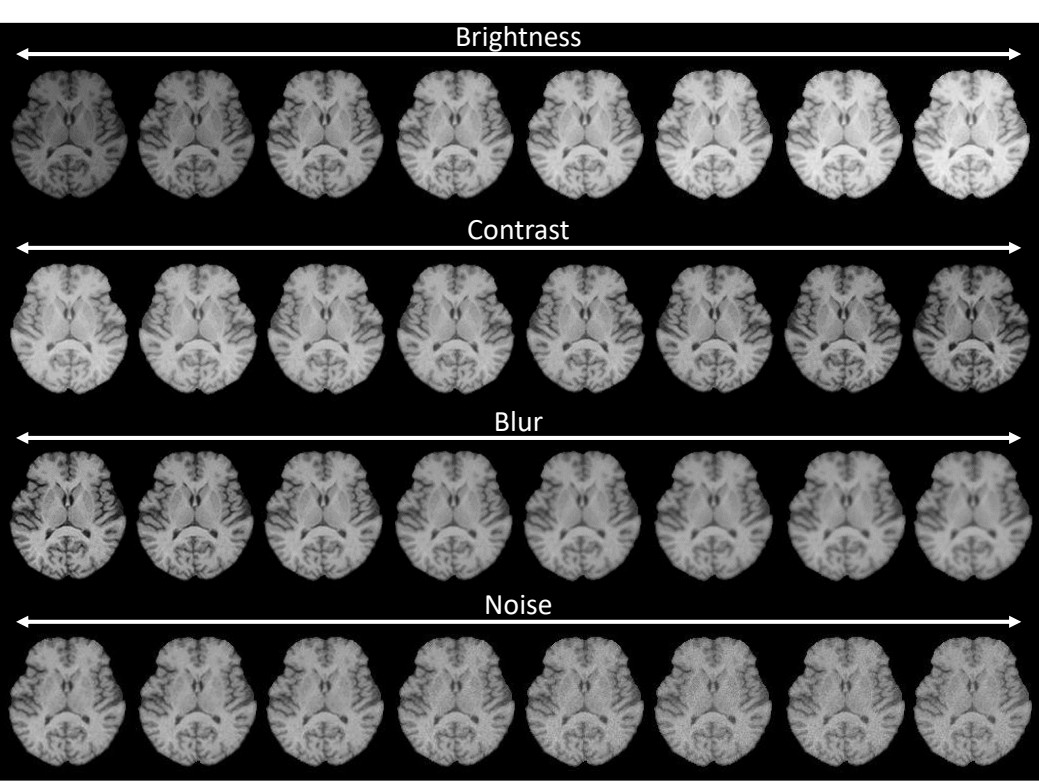

Figure S5: MRI images generated with controlled variations in brightness, contrast, blur, and noise. For each property, a perturbation was applied to the original image to modify only the corresponding attribute. Style vectors were extracted from the perturbed images and interpolated/extrapolated with the original style vector, and combined with a fixed content vector to generate images. The results show that our method can generate a broad spectrum of plausible MRI imaging styles.

## D    ADDITIONAL ANALYSIS OF BAYESIAN OPTIMIZATION

To evaluate the effectiveness of Bayesian optimization, we plotted the black-box model performance (Dice score) over 100 iterations (Fig. S6). The results show that the sampled style vectors led to steadily improved black-box model performance, with reduced variance as the iterations progressed. This indicates that the BO effectively identified high-performing style vectors over time, demonstrating its ability to balance exploration and exploitation. The convergence trend and the discovery of the best-performing sample at iteration 39 further validate the reliability of the optimization process.

This convergence may be attributed to the GP-UCB strategy (Eq. (S7)). This acquisition function is designed to initially explore uncertain regions and gradually shift toward exploitation as predictive uncertainty decreases. Theoretical analysis (Srinivas et al., 2010) shows that the simple regret decays at a rate of $\tilde{\mathcal{O}}(\sqrt{\gamma_T/T})$, where $r_T = f^\star - \max_{t \leq T} f(\mathbf{z}_t)$ and $\gamma_T$ denotes the maximum information gain. This implies that BO can identify near-optimal solutions with a relatively small number of queries, even in high-dimensional settings. In our case, convergence was achieved in fewer than 100 iterations. Additionally, comparison with a random search (Fig. 5) further confirms the superior sample efficiency of BO, highlighting its ability to rapidly focus on high-performing regions.

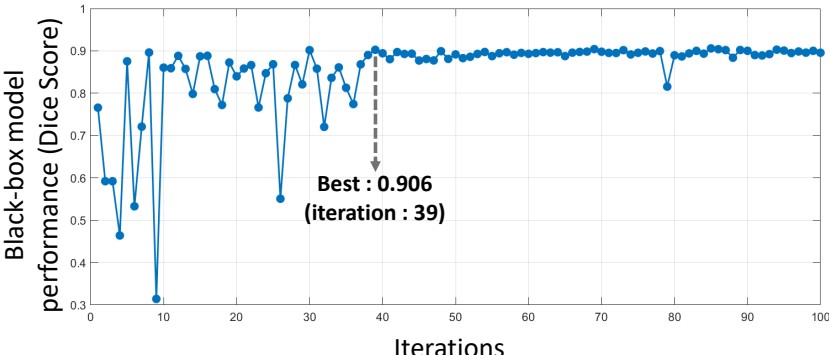

Figure S6: Black-box model performance (Dice Score) over Bayesian optimization(BO) iterations. BO converges toward the source domain, with saturation observed after 39 iterations.

## E    FURTHER EXPERIMENTAL RESULTS

We present additional qualitative results for both harmonization and segmentation performance on all target domains (Domain A, B, C, and D). As illustrated in Figs. S7 and S8, our proposed method successfully harmonizes target images and markedly improves segmentation performance, all without requiring any access to the source domain.

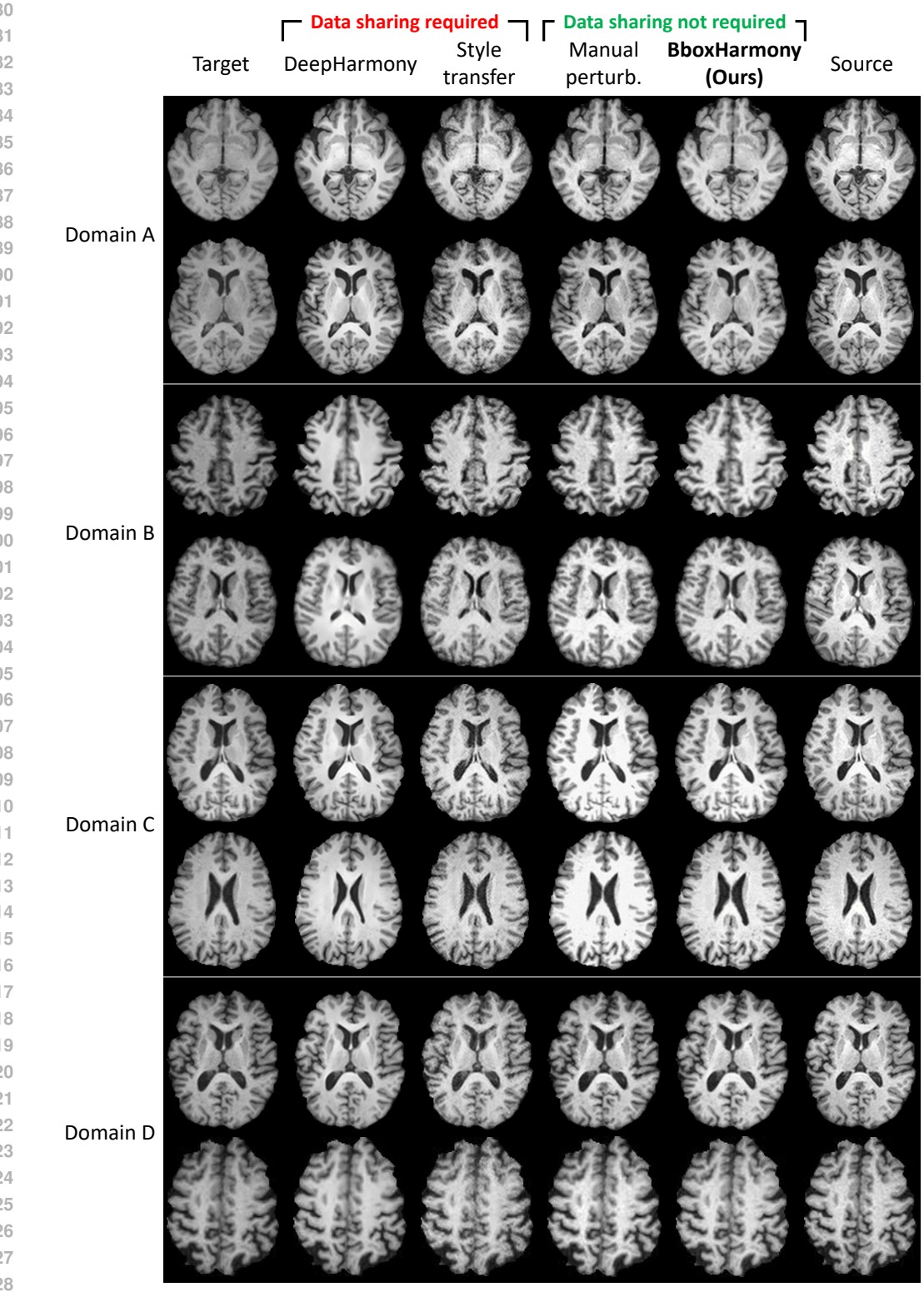

Figure S7: Visual comparison of harmonization results across all four target domains (Domain A, B, C, and D) using different harmonization methods. Methods marked in red require access to source domain data, while those in green do not. BboxHarmony successfully harmonizes target images without requiring any access to the source data.

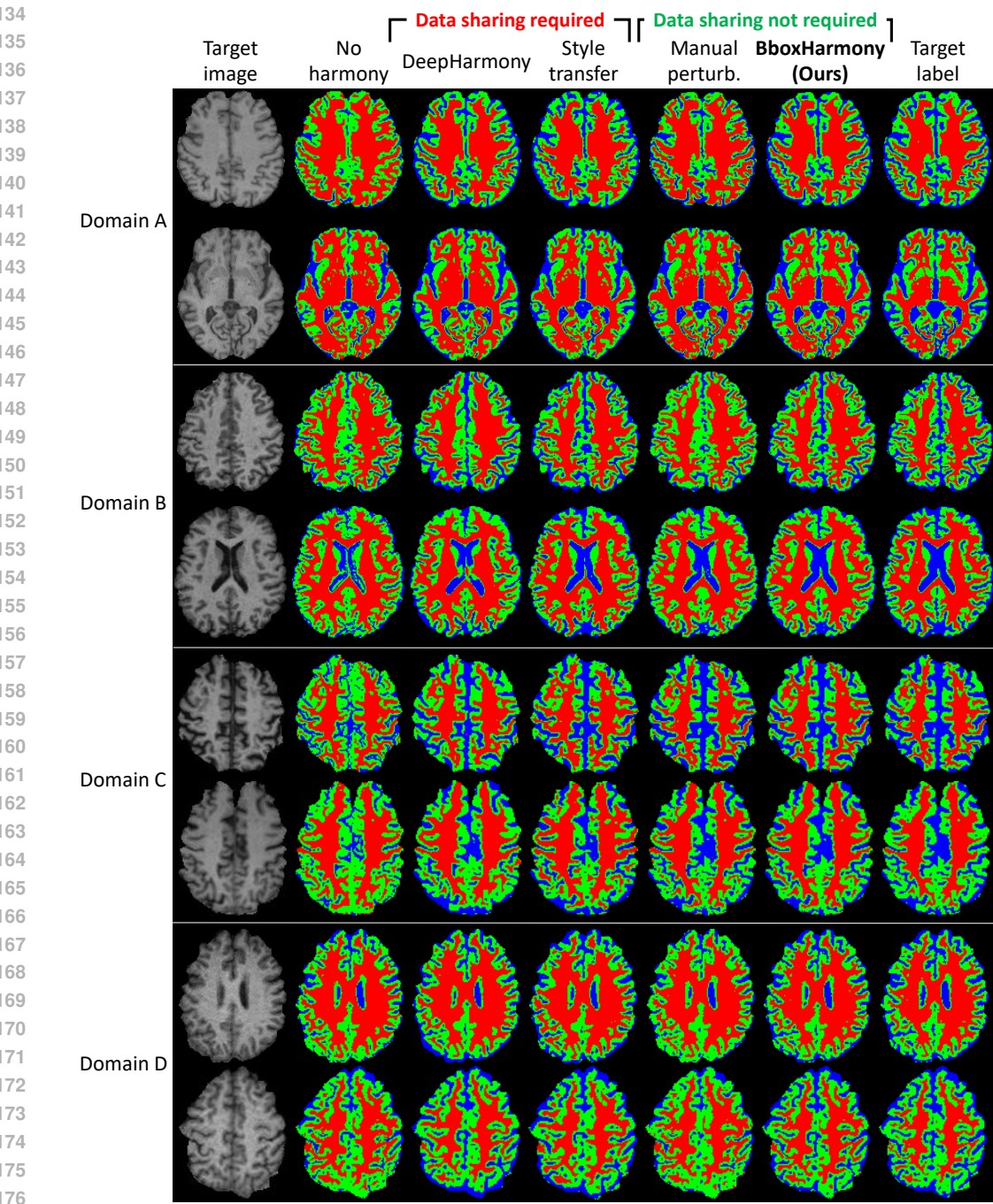

Figure S8: Additional brain tissue segmentation results on target images from all four domains (Domain A, B, C, and D) applying different harmonization methods. Without harmonization (no harmony), performance drops due to domain shift, while harmonization methods generally improve performance. BboxHarmony successfully segments brain tissues, which demonstrates that our method enables the black-box models achieve better performance on an unseen target domain.

# F ADDITIONAL COMPARISON WITH SUPERVISED LEARNING ON LIMITED TARGET LABELS

Although our method does not require any source domain data, it assumes access to a small amount of labeled data from the target domain. In such a scenario, one might question whether simply training a supervised model on the labeled target data could outperform our harmonization approach, especially when the amount of labeled data is sufficiently large. To investigate this, we conducted an experiment comparing our method with fully supervised models trained solely on each target domain. Specifically, we trained U-Net (Ronneberger et al., 2015) models for brain tissue segmentation using varying numbers of labeled subjects from each target domain: 5 (matching our harmonization setting), 10, 20, and 40. The results, summarized in Tab. S3, reveal that when only 5 labeled subjects were available, the supervised models consistently underperformed compared to our method, and competitive performance was only reached after increasing the labeled data, with the required number varying by domain, ranging from 10 to 40 subjects. These results highlight the risk of overfitting with small datasets and the practicality of harmonization in settings where labeled data is scarce. Our method is especially beneficial in clinical environments where obtaining labels typically requires expert knowledge and high costs.

Table S3: Segmentation performance comparison between our method (BBoxHarmony) and fully supervised models trained with varying numbers of labeled subjects (5, 10, 20, and 40) from each target domain. Performance exceeding that of BBoxHarmony is underlined.

| Methods | Domain A | | Domain B | | Domain C | | Domain D | |
|---|---|---|---|---|---|---|---|---|
| | IoU↑ | Dice↑ | IoU↑ | Dice↑ | IoU↑ | Dice↑ | IoU↑ | Dice↑ |
| **BboxHarmony** | 0.830 | 0.906 | 0.825 | 0.902 | 0.805 | 0.884 | 0.830 | 0.905 |
| Supervised model (5 subjects) | 0.669 | 0.790 | 0.758 | 0.841 | 0.546 | 0.684 | 0.727 | 0.838 |
| Supervised model (10 subjects) | 0.838 | 0.911 | 0.763 | 0.861 | 0.715 | 0.825 | 0.814 | 0.895 |
| Supervised model (20 subjects) | 0.838 | 0.911 | 0.798 | 0.882 | 0.771 | 0.862 | 0.828 | 0.903 |
| Supervised model (40 subjects) | 0.844 | 0.914 | 0.854 | 0.916 | 0.813 | 0.889 | 0.870 | 0.928 |

# G GENERALIZATION TO UNSEEN SCANNER VENDORS

To assess cross-vendor robustness, we evaluated harmonization on two vendors (*GE* and *Philips*) unseen during training. The results are summarized in Tab. S4. BboxHarmony consistently improved image similarity (PSNR/SSIM) and downstream performance (IoU/Dice) compared to the non-harmonized inputs. Even without any source-domain data or vendor-specific retraining, BboxHarmony yields sizable gains on unseen scanners, indicating practical deployability in heterogeneous clinical environments.

Table S4: Harmonization on unseen vendor data (PSNR↑/SSIM↑/IoU↑/Dice↑).

| Methods | GE | Philips |
|---|---|---|
| No harmonization | 17.2/0.907/0.553/0.684 | 16.5/0.951/0.514/0.650 |
| **BboxHarmony (ours)** | **18.3/0.926/0.628/0.749** | **21.9/0.954/0.614/0.733** |

# H ADDITIONAL DOWNSTREAM METRICS: SENSITIVITY, SPECIFICITY, AND HAUSDORFF DISTANCE

Beyond IoU/Dice, we report sensitivity, specificity, and 95%-Hausdorff distance (HD) to offer a more comprehensive view of segmentation quality under domain shift. The results are summarized in Tables. S5, S6. Without harmonization, domain shift results in high sensitivity but low specificity. Our method mitigates this imbalance while also reducing boundary errors (HD), improving downstream task performance. BboxHarmony improves specificity and boundary accuracy (HD) across domains, while maintaining strong sensitivity.

Table S5: Quantitative segmentation results using harmonization methods across domains (A,B), reported as *sensitivity↑/specificity↑/HD distance↓*.

| Methods | source data unnecessary | Domain A | | | Domain B | | |
|---|---|---|---|---|---|---|---|
| | | Sens↑ | Spec↑ | HD↓ | Sens↑ | Spec↑ | HD↓ |
| No harmonization | - | $0.903 \pm 0.024$ | $0.944 \pm 0.012$ | $7.82 \pm 1.56$ | $0.991 \pm 0.042$ | $0.956 \pm 0.017$ | $8.39 \pm 2.66$ |
| DeepHarmony | ✗ | $0.956 \pm 0.016$ | $\mathbf{0.963} \pm 0.009$ | $8.43 \pm 1.50$ | $0.859 \pm 0.058$ | $0.955 \pm 0.016$ | $8.54 \pm 1.96$ |
| Style transfer | ✗ | $0.946 \pm 0.014$ | $0.955 \pm 0.010$ | $8.87 \pm 1.81$ | $0.946 \pm 0.030$ | $0.964 \pm 0.011$ | $\mathbf{7.20} \pm 1.37$ |
| BlindHarmony | ✗ | $0.781 \pm 0.100$ | $0.873 \pm 0.025$ | $14.18 \pm 10.39$ | $0.965 \pm 0.059$ | $0.912 \pm 0.040$ | $14.67 \pm 7.64$ |
| Harmonizing flows | ✗ | $0.907 \pm 0.020$ | $0.950 \pm 0.009$ | $7.71 \pm 1.39$ | $0.961 \pm 0.039$ | $0.969 \pm 0.011$ | $7.22 \pm 1.39$ |
| IGUANe | ✗ | $\mathbf{0.971} \pm 0.020$ | $0.957 \pm 0.011$ | $\mathbf{7.38} \pm 1.59$ | $\mathbf{0.989} \pm 0.037$ | $\mathbf{0.970} \pm 0.011$ | $7.94 \pm 1.83$ |
| Manual perturbation | ✓ | $0.945 \pm 0.027$ | $0.952 \pm 0.020$ | $8.31 \pm 2.05$ | $0.973 \pm 0.078$ | $0.970 \pm 0.015$ | $7.04 \pm 2.04$ |
| **BboxHarmony (ours)** | ✓ | $\mathbf{0.986} \pm 0.013$ | $\mathbf{0.965} \pm 0.007$ | $\mathbf{7.21} \pm 1.47$ | $\mathbf{0.991} \pm 0.012$ | $\mathbf{0.973} \pm 0.008$ | $\mathbf{6.76} \pm 1.53$ |

Table S6: Quantitative segmentation results using harmonization methods across domains (C,D), reported as *sensitivity↑/specificity↑/HD distance↓*.

| Methods | source data unnecessary | Domain C | | | Domain D | | |
|---|---|---|---|---|---|---|---|
| | | Sens↑ | Spec↑ | HD↓ | Sens↑ | Spec↑ | HD↓ |
| No harmonization | - | $0.992 \pm 0.032$ | $0.969 \pm 0.011$ | $8.73 \pm 3.51$ | $0.999 \pm 0.006$ | $0.972 \pm 0.008$ | $7.46 \pm 1.48$ |
| DeepHarmony | ✗ | $0.911 \pm 0.018$ | $0.969 \pm 0.011$ | $8.07 \pm 1.46$ | $0.901 \pm 0.041$ | $0.963 \pm 0.007$ | $7.61 \pm 1.31$ |
| Style transfer | ✗ | $0.943 \pm 0.023$ | $0.967 \pm 0.011$ | $8.42 \pm 1.51$ | $0.959 \pm 0.012$ | $0.968 \pm 0.008$ | $\mathbf{7.23} \pm 1.32$ |
| BlindHarmony | ✗ | $0.951 \pm 0.059$ | $0.930 \pm 0.041$ | $16.55 \pm 8.01$ | $0.959 \pm 0.055$ | $0.923 \pm 0.042$ | $14.92 \pm 7.22$ |
| Harmonizing flows | ✗ | $0.955 \pm 0.054$ | $0.974 \pm 0.006$ | $\mathbf{7.90} \pm 1.70$ | $0.980 \pm 0.023$ | $0.972 \pm 0.008$ | $7.42 \pm 1.48$ |
| IGUANe | ✗ | $\mathbf{0.990} \pm 0.017$ | $\mathbf{0.975} \pm 0.009$ | $7.98 \pm 1.96$ | $\mathbf{0.995} \pm 0.009$ | $\mathbf{0.973} \pm 0.007$ | $7.75 \pm 1.54$ |
| Manual perturbation | ✓ | $0.975 \pm 0.045$ | $0.975 \pm 0.009$ | $7.18 \pm 1.91$ | $0.989 \pm 0.016$ | $0.972 \pm 0.009$ | $7.29 \pm 1.68$ |
| **BboxHarmony (ours)** | ✓ | $\mathbf{0.988} \pm 0.020$ | $\mathbf{0.977} \pm 0.007$ | $\mathbf{7.05} \pm 1.60$ | $\mathbf{0.993} \pm 0.009$ | $\mathbf{0.975} \pm 0.007$ | $\mathbf{6.91} \pm 1.49$ |

