# OpenReview forum: "Privacy-Preserving MRI Data Harmonization for Black-box Models"
_ICLR.cc/2026/Conference — ICLR 2026 Conference Withdrawn Submission_

### Official Review · Reviewer_1Lba · 2025-10-29

**Soundness:** 3
**Presentation:** 3
**Contribution:** 3
**Rating:** 4
**Confidence:** 3

**Summary:**

This paper proposes an interesting problem setting and solution for machine learning in the neuroimaging domain. For MRIs, each scanner/domain exhibits some domain shift that is often handled by first harmonizing the images across scanners. This harmonization step typically requires sharing data. Due to privacy constraints with medical data, sharing data to achieve harmonization is often not possible.

The paper therefore proposes to first learn a flexible style transfer model across domains, using public and synthetic data, where each scanner/domain is represented by a style vector. This model can be shared, but a new site must then determine how to set its own target style vector to work effectively with the black-box model. The authors hypothesize that a good target style vector will yield better downstream performance, and empirically support this claim. They propose using Bayesian Optimization (BO) at each site to efficiently search for the best target style vector. Results show that target-domain harmonization and downstream segmentation performance are competitive with methods that rely on explicit data sharing.

**Strengths:**

- Creative problem formulation. The problem definition - harmonizing across domains without sharing data or model parameters - is clearly articulated and motivated by privacy constraints in clinical settings. Framing style discovery as a black-box optimization problem is elegant.

- Methodology. The combination of a disentangled content-style generator and BO over style space is an interesting and reasonable design.

- Evaluation. The experiments test multiple angles: visual quality, traveling-subject consistency, and downstream task performance. Results are competitive with sharing-based baselines despite strict privacy assumptions.

- Acknowledgment of limitations. The paper is careful to discuss scope and limitations, including assumptions about available labels and relatively narrow breadth of evaluation.

**Weaknesses:**

1.	Realism of the black-box setting.
While privacy restrictions in healthcare are real, the precise scenario envisioned here is somewhat hard to map to practice. If the model provider sends a pre-compiled model that hospitals can run but not fine-tune, that would satisfy the “black box” constraint, but I personally have never heard of this. Conversely, if the model is accessed via an API, then the hospital must transmit private data externally, which seems to violate the privacy constraint motivating the setup. Clarifying how this setting might appear in practice would strengthen the motivation.
2. Is the black-box model itself trained on data that has been harmonized using the same (or similar) method? If the model was trained to perform robustly across domains from the start (without harmonization), couldn't it reduce the signal available for the BO procedure? More broadly, the efficacy of your method could depend on unknown training choices for the black box model. If we are going to require the blackbox trainer to do certain things, then maybe we can bypass this whole procedure, as discussed in the next point.
3. Simpler alternatives to consider / contrast?
Wouldn’t it be much simpler if the black-box model provider simply shared their style vector, or if all sites harmonized toward a public reference style vector (e.g., from a public dataset)? This might preserve privacy while avoiding the need for per-site BO optimization. Arguments for why this simpler solution might be infeasible would strengthen the significance of the method.
4.	Correlation analysis.
The observed correlation between style similarity and performance (Fig. 2) supports the intuition but the correlation appears modest. The points in Fig. 2 are from randomly sampled domains, but when we start optimizing the style vector, the relationship may not behave in the same way. The experiments somewhat mitigate this concern, but additional validation would help. I was wondering, for instance, whether BO approaches the "true" target vector or whether there are many target vectors which give good performance. (Maybe the best style vector is actually totally different from the ground truth. It could be that what you are actually accomplishing here is a backdoor way to fine-tune the model through preprocessing!)
5.	Notation clarity.
The notation is a bit confusing,  I think. For instance, compare Eq. 2 to Eq. 5. Eq. 2 makes it look like we can work with an individual instance, but Eq. 5 (logically, I think) is defined in terms of performance on a set. When I saw Eq. 2 I was worried that the results in Fig. 2, which occur on many samples in a domain, would not apply when we optimize on *instances*. Luckily that is not what is done. Still, I think this presentation could be clarified. (You use few instances in practice, 5, but because it is a segmentation task, you actually use many labels. I would assume that this is important for the method to work.)
6.	Applicability to other models and tasks.
The segmentation task is ideal for this setup, since the dense labels provide a strong signal for BO. It’s less clear whether this method would work for classification or regression tasks where per-instance feedback may not be as informative about the style vector. That would add another dependency on the blackbox model if the approach can only work on certain blackbox tasks.

7. Other related work.
I think there are a lot of other approaches that could satisfy the black box/private problem setting here. "Domain generalization" methods, in particular, I think could be competitive, where we train a model (either the black box or harmonization) on some sites, but it can generalize to unseen sites. I think it would strengthen the paper to compare to these ideas. A few examples in a quick search (maybe not directly relevant, but just to give the idea).
- Contrastive Anatomy-Contrast Disentanglement: A Domain-General MRI Harmonization Method, MICCAI 2025
- FedHarmony (MICCAI 2022), only shares some feature statistics I think (but more targeted to federated setting).
- SFHarmony: Source Free Domain Adaptation for Distributed Neuroimaging Analysis, CVPR 2023

**Questions:**

My main question is whether the authors can comment on how realistic the setting is, and feasibility of alternatives, as discussed in W1-W3, W7?


Some minor questions:
- The method relies on a small but labeled target set (five subjects in experiments). This could hinder adoption in medical settings, where a more "turnkey" approach would be desired.
- How are the target-set samples for BO chosen? Are they disjoint from the validation/evaluation sets?
- Could the approach be relevant to a federated learning setup, where each site optimizes locally and shares only aggregated signals?
- Could gradient-based optimization on the style vector (e.g., zeroth-order gradient descent) be viable, or does that break the black-box assumption?

---

### Official Review · Reviewer_wGmU · 2025-11-01

**Soundness:** 2
**Presentation:** 3
**Contribution:** 2
**Rating:** 4
**Confidence:** 4

**Summary:**

The paper proposes a privacy-preserving approach to mitigate MRI domain shifts arising from different scanners, acquisition parameters, and other data-collection artifacts. It learns source-target harmonization under label scarcity and provides some evidence consistent with preserving anatomically relevant structures for segmentation tasks. The authors report improvements across multiple metrics and provide evidence for their key assumption that black-box model performance can guide the search over a learned style-only latent space.

**Strengths:**

-Cross-vendor generalization: authors evaluate performance on unseen manufacturers (GE, Philips) and still improve PSNR/SSIM/IoU/Dice over no-harmonization, supporting practical deployability beyond same-vendor settings.

-Competitive under label scarcity: with only 5 labeled target subjects, their approach outperforms target-only supervised U-Nets.

-Style coverage evidence: a domain-classifier + t-SNE shows generated images spreading across domain clusters.

**Weaknesses:**

Relevance: authors mention that “in many clinical settings under strict regulation [...] models are often deployed as privacy-preserving black-box”. Please provide concrete deployments or citations beyond regulations, so readers see real demand for this setting.

Quality
A central assumption is that a black-box model performance (Eq. 5) is enough to guide style exploration to match the one in the source domain, without harming anatomy (content). However, the performance of the black-box model is directly its segmentation. Under a different task, e.g. classification, it is unclear if black-box performance is enough to guide the optimization without exploiting style errors that align with the source model biases.
Authors test performance under a “Manual Perturbation” setting, but should show that using the same data preprocessing protocol for target and source (e.g. statistical normalization) is not enough for harmonizing data from different domains under their proposed metrics.
Authors assess performance for different MRI vendors (GE/Philips), but only report performance for their harmonization method vs no harmonization. They should compare BboxHarmony against the other harmonization methods as well, especially the Manual Perturbation setting, which respects their privacy constraints.

Missing references to relevant works including

[1] Van der Goten L, Hepp T, Akata Z, Smith K. Conditional De-identification of 3D Magnetic Resonance Images. British Machine Vision Conference (2021).
[2] Van der Goten, L. A., & Smith, K. (2024). Privacy Protection in MRI Scans Using 3D Masked Autoencoders. In International Conference on Medical Image Computing and Computer-Assisted Intervention (MICCAI) (2024), 583-592.

**Questions:**

In the qualitative comparisons (e.g., Fig. 6), IGUANe is not shown, yet Table 3 reports it as the second-best method. Could you explain why it was omitted?
For the supervised fine-tuning baseline on the target domain (Appendix F), was the network identical to the source black-box model (especially architecture, capacity, input preprocessing, optimizer, and training budget/epochs)? If the supervised target baseline uses a different configuration, its performance is not directly comparable to the black-box + harmonization pipeline.
Missing citation/definition for SSIM, IoU and PSNR.

---

### Official Review · Reviewer_KwfZ · 2025-11-03

**Soundness:** 2
**Presentation:** 1
**Contribution:** 2
**Rating:** 2
**Confidence:** 3

**Summary:**

This work explored privacy-preserving MRI harmonization (conditional image translation/generation) by proxying the domain shifts using a segmentation network. A segmentation network was trained on one source domain, and a conditional GAN (with content and style encoders) was trained on multiple target domains using disentanglement learning. A Gaussian process was applied to estimate the style vector that makes the translated target image similar to the source image (proxyed by the segmentation performance of the translated target image). The proposed framework was evaluated on 5 Siemens scanners, comparing against other comparison methods, and on 2 other vendor scanners with no harmonization baseline. The proposed methods achieved the best segmentation and similarity metrics in the experiment.

**Strengths:**

1. Privacy-preserving data harmonization is an important and challenging question. However, there are multiple points/implementations in the manuscript that raise questions about whether data and model sharing are rigorously prevented in the proposed framework, or if the framework employs some unrealistic and impractical assumptions.

2. Proxying the domain shifts using segmentation performance is a novel idea. Although the reviewer is unsure about how to get the segmentation performance when the new data is not co-registered with the source data (which requires data sharing).

**Weaknesses:**

1. The biggest weakness of this paper is that the datasets included in the current manuscript do not seem to have large domain shifts and visual discrepancies. From Fig. 6 and Fig. S2, the domain shifts are not as severe as those studied in Harmonizing Flow. For example, in Fig.8 of Harmonizing Flow, there is a significant difference between the dHCP and V2LP datasets. The better performance in the easy case is not necessarily transferable to the hard, more challenging scenario. Therefore, this is a major drawback of the current manuscript.

2. I don't understand why the model parameter accessibility and data sharing is not required for this work. Assuming that hospital A developed the segmentation model on their dataset, and hospital B wants to use that to segment its own data. Because of the privacy concern, B can not transfer data to A. To train BBoxHarmony, A still needs to provide the weights of segmentation models to B, so B can reuse the segmentation. Additionally, according to the provided code, the BO process requires all source data, source labels, and target data. Which means B needs to ask for source data from A to train BBoxHarmony. How does that prevent data sharing?

3. There are many hyperparameters that are not explicitly described/introduced in the main text, such as the dimension of the style vector (32), the 100 random samplings of the style vectors. These parameters influence both generation and BO. How are they determined? A sensitivity analysis could be beneficial.

4. The presentation is not good enough. Many technical details remain unclear. Please refer the the questions below.

**Questions:**

1. For Fig. 2. c, as indicated in line 203, when image similarity between source and target domain images was calculated, were they co-registered? Is there any explanation or qualitative examples of those outliers (e.g., low similarity with ~0.8 DSC and high similarity with ~0.45 DSC)? Throughout the paper, when SSIM/pSNR is measured, are they calculated on paired/co-registered scans only?

2. Fig. 3, the location of each image should be left, middle, and right, not top, middle, and bottom.

3. Fig. 5, given the y-axis of these figures, it seems the improvement brought by both Bayesian optimization and random search is negligible (i.e., DSC from 0.90 to 0.906, SSIM from 0.928 to 0.934). What is the point of these marginal improvements?

4. Is the search for style vector only used 50 slices/scan * 5 scans = 250 slices?

5. In Fig. 6, the Source Domain A and Domain B seem to have exactly the same contrast and do not have domain shifts. The difference between Target Domain A (B) and Source Domain A (B) is just brightness, which could be easily adjusted by percentile normalization. What is the domain shift that these datasets cover? Figure 8 of Harmonizing Flows covers two domains with significantly different appearances and contrasts (dHCP and V2LP). Is the proposed framework tested on domains with very significant discrepancies, just as Harmonizing Flow did?

6. The notation P is often used for probability (distribution), while this paper used it for performance. Given it also introduced Bayesian optimization, this notation (P as performance) is a bit confusing.

7. What is the reason for only providing visualization of DeepHarmoney, Style Transfer, and manual perturbation? Why not provide visualizations of Harmonizing flows and IGUANe? IGUANe provides the second-best downstream task performance, while BlindHarmony has the worst performance. Their visualization is absolutely more necessary than the other two.

8. Based on Line 215 of the supplemented code train2_BO.py, to calculate the black-box performance during the BO process, it assumes the target image and source image are co-registered.
 - `score = f1_score_multi(bbox_output, bbox_label.squeeze(1), num_classes=opts.n_classes)[1:].mean()`
If this is the correct understanding, what would happen if the income image is from an unseen domain and is not registered with the source image? In that case, isn't the data sharing necessary to co-register images from source and target domains?

---

### Author Response · Authors · 2025-11-20

We sincerely appreciate the reviewers’ constructive comments and thoughtful feedback.
After careful consideration, we have decided to withdraw the manuscript in order to address the highlighted issues and substantially improve the quality of the work. We look forward to resubmitting a strengthened version in the future. Thank you.

---

### Note · Authors · 2025-11-20

I have read and agree with the venue's withdrawal policy on behalf of myself and my co-authors.